# Antimicrobial Resistance Patterns and Risk Factors Associated with *Salmonella* spp. Isolates from Poultry Farms in the East Coast of Peninsular Malaysia: A Cross-Sectional Study

**DOI:** 10.3390/pathogens10091160

**Published:** 2021-09-09

**Authors:** Abdinasir Yusuf Osman, Sharifo Ali Elmi, David Simons, Linzy Elton, Najmul Haider, Mohd Azam Khan, Iekhsan Othman, Alimuddin Zumla, David McCoy, Richard Kock

**Affiliations:** 1The Royal Veterinary College, University of London, Hawkshead Lane, North Mymms, Hatfield AL9 7TA, UK; dsimons19@rvc.ac.uk (D.S.); nhaider@rvc.ac.uk (N.H.); rkock@rvc.ac.uk (R.K.); 2Faculty of Veterinary Medicine, Universiti Malaysia Kelantan, Pengkalan Chepa, Kota Bharu 16100, Malaysia; sharifo.d18d006f@siswa.umk.edu.my (S.A.E.); azamkhan@umk.edu.my (M.A.K.); 3Centre for Clinical Microbiology, Department of Infection, Division of Infection and Immunity, University College London, London NW3 2PF, UK; linzy.elton@ucl.ac.uk (L.E.); a.zumla@ucl.ac.uk (A.Z.); 4Jeffrey Cheah School of Medicine and Health Sciences, Monash University Malaysia, Jalan Lagoon Selatan, Bandar Sunway 46150, Malaysia; iekhsan.othman@monash.edu; 5National Institute for Health Research Biomedical Research Centre, University College London Hospitals, London NW1 2BU, UK; 6Institute of Population Health Sciences, Barts and London Medical and Dental School, Queen Mary University of London, London E1 2AD, UK; d.mccoy@qmul.ac.uk

**Keywords:** *Salmonella* spp., antimicrobial resistant, distribution, poultry farms, environment, Malaysia

## Abstract

The burden of antimicrobial use in agricultural settings is one of the greatest challenges facing global health and food security in the modern era. Malaysian poultry operations are a relevant but understudied component of epidemiology of antimicrobial resistance. We aimed to identify the prevalence, resistance patterns, and risk factors associated with *Salmonella* isolates from poultry farms in three states of East Coast Peninsular Malaysia. Between 8 February 2019 and 23 February 2020, a total of 371 samples (cloacal swabs = 259; faecal = 84; Sewage = 14, Tap water = 14) was collected from poultry operations. Characteristics of the sampled farms and associated risk factors were obtained using semi-structured questionnaires. Presumptive *Salmonella* spp. isolates were identified based on colony morphology with subsequent biochemical and PCR confirmation. Susceptibility of isolates was tested against a panel of 12 antimicrobials using disk diffusion method. Our findings revealed that the proportion of *Salmonella* spp.-positive isolates across sample source were as following: cloacal swab (46.3%, 120/259); faecal (59.5%, 50/84); in tap water (14.3%, 2/14); and in sewage sample (35.7%, 5/14). Isolates from faecal (15.5%, 13/84), cloacal (1.2%, 3/259), and sewage (7.1%, 1/14) samples were significantly resistant to at least five classes of antimicrobials. Resistance to Sulfonamides class (52%, 92/177) was predominantly observed followed by tetracycline (39.5%, 70/177) and aminoglycosides (35.6%, 63/177). Multivariate regression analysis identified intensive management system (OR = 1.55, 95% CI = 1.00–2.40) as a leading driver of antimicrobial resistance (AMR) acquisition. A prevalence of resistance to common antimicrobials was recorded for sulfamethoxazole (33.9%), tetracycline (39.5%), and trimethoprim-sulphamethoxazole (37.9%). A close association between different risk factors and the prevalence of AMR of *Salmonella* strains suggests a concern over rising misuse of veterinary antimicrobials that may contribute to the emergence and evolution of multidrug-resistant pathogen isolates. One Health approach is recommended to achieve a positive health outcome for all species.

## 1. Introduction

Antimicrobial resistance is one of the biggest threats to global health and food security; today, this is rising globally in both developed and developing countries. Antimicrobial resistance occurs naturally, or it can be acquired by bacteria. Antimicrobial selective pressure due to inappropriate and overuse of antibiotics may promote the emergence of the phenomenon. Cross-sectoral interconnectivity through healthcare, agriculture, and environment contributes further emergence, evolution, and global spread of antimicrobial resistance [1]. The rise in multidrug-resistant (MDR) bacterial infections is being driven by the global expansion of livestock production systems where antimicrobials are used routinely to maintain livestock health and productivity. In low- and middle-income countries (LMICs), 73% of all antimicrobials are used in animals raised for food [2]. More specifically, in Malaysia and many other Southeast Asian countries, a wide use of antibiotics, especially in intensive production system, is linked to the higher resistance to various antibiotics [3]. Most of these antimicrobial compounds are accumulated and biomagnified through the food chain. Exposure among human populations to low levels of antimicrobial contaminants through marine and agricultural ecosystems has been linked to development and acquisition of antibiotic-resistant bacteria [1,4].

*Salmonella* spp. are the cause of one of the most common bacterial infections in humans. A substantial number of pathogenic strains of *Salmonella* spp. cause food-related poisoning worldwide [5]. The global burden of non-typhoidal *Salmonella* spp. (NTS) is increasing, with over 94 million cases of gastroenteritis, which is responsible for 77,500 in 2017 [6]. The development of antimicrobial drug resistance in non-typhoidal *Salmonella* spp. is an almost inevitable consequence of the use of antibiotics in animal husbandry [7]. Practices such as rampant use of broad-spectrum antimicrobials administered in low doses for growth promotion and use of non-approved drugs or drugs used in off-label scenarios are driving the emergence of antimicrobial resistance in veterinary settings [8]. Of particular concern is the development of resistance to key antibiotics, such as the fluoroquinolones [9], β-lactams, and colistin [10].

Malaysia is among the top consumers of poultry meat worldwide, and the scale-up and intensification of poultry farming has led to the steady rise of antimicrobial-resistant *Salmonella* spp. infections [8]. Poultry make up the largest share of livestock in Malaysia [11]. 

The reliance on antimicrobials to meet demand for animal protein poses a serious public health consequence and a likely threat to the sustainability of the livestock industry and thus to the livelihood of farmers [12]. Effectivity and scalability of AMR surveillance with the recognition of One Health approach as center of governance is appreciated worldwide [13]. However, epidemiological investigations for proper understanding of the context and assessment of the ultimate and proximate drivers of AMR are poorly documented in the east coast of peninsular Malaysia. In the absence of systematic surveillance systems, the use of point prevalence surveys in these operations represent a largely untapped source of information to map trends in AMR.

We present findings from a study of risk factors associated with the carriage of resistant *Salmonella* spp. isolates in poultry farms of East Coast Peninsular Malaysia to establish a baseline for monitoring AMR levels in these settings for policy makers.

## 2. Results

We administered a semi-structured questionnaire to 31 poultry farmers and conveniently sampled 14 farms with a total of 371 samples across three states of peninsular Malaysia. The 14 included poultry farms were in Kelantan, Terengganu, and Pahang, located in east coast of peninsular Malaysia. The socio-demographic traits of the included farms is given in Appendix A. Of these, 371 samples (cloacal swabs = 259; faecal samples = 84; sewage = 14, tap water = 14) were collected from 14 poultry farms: 158 from Kelantan, 80 from Terengganu, and 133 from Pahang. Of the tested 371 samples, 177 (47.7%) were *Salmonella* spp. positive (Table 1). Univariate analyses identified several variables significantly associated with *Salmonella* spp. positivity (*p* < 0.05), such as sample source, location, sewage system, and water source. Most of these variables are poultry-farm-contact related. The proportion of *Salmonella* spp.-positive isolates across sample source were as follows: cloacal swab (46.3%, 120/259); faecal samples (59.5%, 50/84); in tap water (14.3%, 2/14); and in sewage sample (35.7%, 5/14). The proportion of *Salmonella* spp.-positive isolates among the states were not significantly different (*p* > 0.065) (Table 1).

Among the districts, the highest prevalence of *Salmonella* spp. was recorded in Kuantan farms (63.3%, 50/79) followed by Machang (57.1%, 16/28) and Bachok (55.8%, 29/52) and Pasir Mas (50% 13/26), respectively (Table 1). The proportion of *Salmonella* spp.-positive isolates among water source are as follows: surface water (35.8%, 38/106), bond water (54.1%, 72/133), and pump water (50.8%, 67/132) (Table 1). We observed that 86.4% of the *Salmonella* spp. isolates were resistant to the tested panel of antimicrobials, and MDR strains were 41.2% (Table 2).

Resistance to sulfonamides class (52%, 92/177) was predominantly observed followed by tetracycline (39.5%, 70/177) and aminoglycosides (35.6%, 63/177), whereas chloramphenicol (7.9%, 14/177) and cephalosporins (11.9%, 21/177) were the least resistant classes for the isolated *Salmonella* spp. Figure 1 shows the prevalence of antimicrobial class-resistant *Salmonella* spp. isolated from poultry farms collected from Kelantan, Terengganu, and Pahang poultry operations. We observed the resistance patterns of *Salmonella* spp. isolates against a panel of 12 antimicrobials were similar across the participated states. However, the prevalence of resistance to trime-thoprim-sulphamethoxazole was consistently higher than other tested antimicrobials (Figure 2). Similarly, the highest resistance was noted in faecal samples, followed by cloacal and sewage systems (Figure 3). The source of the sample, production system, management system, the size of the farm, poultry origin, and source of the water factors were significantly associated with at least one antimicrobial (Appendix A). Furthermore, we observed that isolates collected from faecal (15.5%, 13/84), cloacal (1.2%, 3/259), and sewage (7.1%, 1/14) samples were significantly resistant to at least five classes of antimicrobials, whereas surface water (4.9%, 5/103), bond water (12%, 16/133), and pump water (9.1%, 12/132) were significantly resistant to at least two classes of antimicrobials (Table 3). The multivariate regression analysis for the management system of the farms with special-reference intensive farms (OR = 1.55, 95% CI = 1.0–2.4) were significant (*p* < 0.05) as leading drivers of *Salmonella* spp. antimicrobial resistance in the participating states of East Coast Peninsular Malaysia (Table 4).

For PCR analysis, eight resistance genes, including *bla*_TEM_ for β-Lactams, *tet* (*A*) and *tet* (*B*) for tetracyclines; *catA1, cat2,* and *floR* for chloramphenicol; and *sul1* and *sul2* for sulfonamides, were identified in the tested *Salmonella* spp. isolates (Table 5). Among them, 85%, were found to harbor *sul1*, followed by *cat2* 78%, *floR* 78%, *sul2* 71%, and *bla*_TEM_ 42%.

## 3. Discussion

This study aimed to identify the prevalence, resistance patterns, and risk factors associated with *Salmonella* spp. resistance from poultry farms in Kelantan, Terengganu, and Pahang states of East Coast Peninsular Malaysia. The results suggested farm-contact related variables, including sample source, location, sewage system, and water source, were significantly (*p* < 0.05) associated with *Salmonella* spp. positivity. The findings are comparable to studies from Peninsular Malaysia [14], Indonesia [15], Thailand [16], and Vietnam [17], all of which reported high levels of *Salmonella* spp. prevalence. This indicates that contamination by *Salmonella* in these farms greatly increases the risk of human exposure and the need for improved monitoring and surveillance systems that address environmental sanitation and behavioral intervention. Notably, the results also revealed *Salmonella* spp. isolates were resistant to most of the antimicrobials tested, with special reference to tetracycline, sulfamethoxazole/trimethoprim, sulfamethoxazole, gentamicin, and ampicillin [18]. These resistance rates reflect their widespread use and are consistent with similar studies that report a minimum level of resistance against the tested panel of antibiotics in poultry settings [19,20]. The fact that our study found low resistant levels against amoxicillin and ciprofloxacinreflects is possibly because these antibiotics are not used for therapeutic purposes in clinical veterinary medicine in Malaysia [21]. Notably, the antimicrobial drugs, including tetracyclines and sulfonamides, are most commonly used in farm animals to promote growth production. In this study, we observed that the resistance patterns of *Salmonella* spp. isolates against a panel of 12 antimicrobials are generally similar in all selected states of East Coast Peninsular Malaysia that include Kelantan, Terengganu, and Pahang. However, there is considerable variation in the prevalence of *Salmonella* spp. resistance between districts. These differences are associated to farm-specific risk factors. The prevalence of *Salmonella* spp. resistance to tetracycline, sulfamethoxazole/trimethoprim, sulfamethoxazole, gentamicin, and ampicillin was consistent in all three participating states. This resistance also reflects the common use of antimicrobials in these poultry operations as well as in other agricultural activities [12]. Moreover, most of these antimicrobials are also used in human medicine, with special reference to tetracycline, sulfamethoxazole, and ampicillin [22]. Our results are consistent with those of other studies across peninsular Malaysia. For example, chicken flock sampling in south-central peninsular Malaysia found that *Salmonella* spp. were resistant to ampicillin (17.6%), tetracycline and streptomycin (35.3%), sulfonamides (29.4%), trimethoprim (20.6%), nalidixic acid and colistin (14.7%), chloramphenicol and nitrofurantoin (11.7%), amoxicillin-clavulanate (5.9%), kanamycin and cefotaxime (2.9%), gentamicin, ciprofloxacin, norfloxacin, and ceftiofur (0%) [23]. 

Of note, our farm-level estimates are based on non-randomly selected samples, and we should expect these estimates to be different than estimates from randomly selected samples. For example, in *Salmonella* spp., high percentages of resistance were found, such as to sulphonamide (96.5%), ampicillin (89.5%), tetracycline (85.1%), chloramphenicol (75.4%), trimethoprim (68.4%), trimethoprim-sulfamethoxazole (67.5%), streptomycin (58.8%), and nalidixic acid (44.4%) [24].

Implementation of biosecurity levels including improved sewage systems, personal protective equipment (PPE), washing facilities, use of disinfectant, and source of the food were not important factors for the occurrence of *Salmonella* spp. and AMR in the sampled poultry farms. In this study, the majority of farmers reported antimicrobial usage for prophylactic, treatment, and productivity purposes [25]. This reflects substandard farm management conditions in which poultry disease frequently occur along with global expansion of intensifications. Furthermore, the cost associated with veterinary services, including treatment and laboratory diagnostics, might further exacerbate the misuse of antimicrobials [12]. While the ban of antibiotic growth promoter has been globally implemented including EU countries [26,27,28,29,30], circumstantial evidence suggest their use in farm-produced animals in South East Asia, including Malaysia [31,32]. Little data on awareness campaigns of antimicrobials usage exist so far in the livestock-production system across South East Asia, including Malaysia [31,33,34].

Regarding PCR analysis, most isolates harbored *sul1*-resistant genes (85%), followed by *cat2* (78%), *floR* (78%), *sul2* (71%), and *bla*_TEM_ (42%). This was in agreement with other results of previous studies reported in South East Asia [35] and China [36]. Difference in the distribution of resistance genes across tested strains remains unclear. However, the high frequency of resistant genes reflects resistance to sulfonamides along with co-selection factors in poultry *Salmonella* spp. 

These findings were comparable to our previous study in which poultry *E. coli* isolated harbored *sul1*-resistant genes (100%) [8]. 

Furthermore, the prevalence of *Salmonella* spp. resistance in source samples, sewage, and water sources was significantly (*p* < 0.005) associated with AMR acquisition. Importantly, most of the risk factors were associated with resistance to at least one antimicrobial agent (Appendix A). Notably, intensive management systems (OR = 1.55, 95% CI = 1.0–2.4) had an increased frequency of AMR, as the agricultural intensive farming systems have long been recognized as hotspots of drug resistance in low- and middle-income countries (LMICs) in South East Asia [37]. This indicates the necessity of a transition to sustainable animal production in Malaysia in which government enhances the farm-level biosafety and biosecurity [38]. The resistance patterns found in the cloacal, faecal, sewage, and tap samples in this study have been found to be similar to those reported in clinical-based surveillance studies [39]. The lower prevalence of resistance in sewage and tap water isolates, however, could be correlated with sensitivity, as it is likely lower than isolate-based surveillance [40]. The source of water and the presence of a sewage system were identified as important risk factors for the presence of AMR in *Salmonella* isolates in the study sites. Importantly, the sampled poultry farms usually access drinking water from intact sources, and thus the association could reflect contact transmission at the farm level. This association has important implications for low-income countries, where potable water remains a pressing challenge [41]. Consumption of poultry meat and its products is increasing, and most poultry meat and eggs are produced and distributed through informal sources that operate outside national quality-control standards and regulations [42,43]. Generally, poor sewage systems along with presence of manure and rubbish from these operations increase the likelihood of multidrug-resistant *Salmonella* spp. carriage in synanthropic wildlife, which in return galvanizes the dissemination of clinically relevant AMR between sympatric wildlife, humans, livestock, and their shared environment. These associations were more pronounced for seed-eating birds and wild boars across different urban ecological systems [44,45]. This denotes the pressing need to effectively enforce environmental legislation and unregulated antibiotic use in agricultural setting. In the absence of national systematic surveillance, our point prevalence surveys of AMR in poultry farms of East Coast Peninsular Malaysia are useful to guide potential future interventions of AMR. The close association between different risk factors and the high prevalence of resistant in *Salmonella* strains indicates increased exposure to antimicrobials and suggests a concern over rising misuse of veterinary antimicrobials that may contribute to the emergence and evolution of multidrug-resistant pathogen isolates. Public health interventions to limit AMR need to be tailored to local poultry-farm practices that affect bacterial transmission. Cross-sectoral collaboration and enhancement of surveillance systems, including developing alert mechanisms for early detection and reporting of AMR, will drive improved policy formulation and its translation into effective implementation. Improving certain domains, including public awareness and education, antimicrobial stewardship and medicines regulation, as well as AMR research and fostering implementation research using One Health approach, is recommended.

## 4. Materials and Methods

### 4.1. Ethics Approval

This study was approved by the Institutional Research Ethics Committee of the Faculty of Veterinary Medicine, University Malaysia Kelantan (UMK) (Ref: 12/2018).

### 4.2. Study Design and Data Sources

We performed a cross-sectional study targeting poultry farms in three states of East Coast Peninsular Malaysia that include Kelantan, Terengganu, and Pahang (Figure 4). Figure 5 depicts the study organizational chart of sites, farms, risk factors, and flow of sample collection, laboratory processing, and analyses by antibiotic class.

### 4.3. Data and Sample Collection

A total of 371 samples (cloacal swabs = 259; faecal = 84; sewage = 14; tap water = 14) were collected between 8 February 2019 and 23 February 2020. Data pertaining to farm characteristics, including management, biosecurity, and disease history along with antimicrobial usage, were collected using semi-structured questionnaires. A total of 31 farmers that met inclusion criteria of keeping poultry farms and who responded with written consent were included in the analyses in (Appendix A). Regarding the management system, flock size, and sewage system, the following definitions and criteria were used:

Intensive management system is defined as mainly concentrated and often mechanized operations that use controlled-environment systems to provide the ideal thermal environment for the poultry.

Semi-intensive system is that which relies on natural airflow though the shed for ventilation.

Extensive system is mainly pasture-based and land-based, where birds in the household flock are typically housed overnight in the shelter and are let out in the morning to forage during the day.

The criteria of the farm size included large-scale commercial farms that have ≥10,000 birds, medium-scale commercial farms that have 5000–10,000, and small-scale farms where birds are often kept in single-age groups of >1000.

A poor sewage system is defined as one that retains high volumes of wastewater with low flow rate, blackish appearance, and sewage smell odour as a result of composing agricultural waste—probably as leakage from nearby irrigated effluent that is used for agricultural land application along with the presence of food waste, green waste, plastic, and heavy materials.

A good sewage system is one that has good drainage with no agricultural waste and relatively low heavy materials.

Excellent sewage system is one that has significant drainage, no agriculture, and no heavy materials.

### 4.4. Samples Collection and Laboratory Methods

The samples were collected according to standard operating procedures and good laboratory practices. Briefly, the cloacal swab samples were collected using sterile transport media; faecal samples using sterile containers and water samples using sterile water bottles were kept in a cooling box containing ice bags, maintaining low temperature at (4 °C) before transferring to the clinical laboratory within 24–48 h for pathogen culturing. All cloacal swabs and fresh faecal samples were placed in Amies transport media and transported on ice to the molecular biology laboratory, University Malaysia, Kelantan (UMK). Sewage and tap water samples were transported in conical tubes, all on ice. The number of samples per farm is given in Appendix A.

### 4.5. Microbiological Testing

Samples were enriched in buffered peptone water for 24 h at 37 °C, and then pre-enriched 0.1 mL and 1 mL cultures were incubated in 9.9 mL of Rappaport Vassiliadis Soy Broth (RVS) at 42 °C and 9 mL of Muller–Kauffmann Tetrathionate–Novobiocin (MKTTn) broth at 37 °C for 24 h, respectively. Loopfuls of RVS and MKTTn cultures were streaked onto selective agar plates (Brilliant Green Agar (BGA)) and then incubated for about 24 h at 37 °C. Suspected *Salmonella* spp. colonies were picked from each plate, purified, and subjected to biochemical tests (Appendix A). All media used were purchased from Oxoid, Basingstoke, Hampshire, UK. Cultured bacteria were routinely stored with 20% of glycerol stock at −20 °C and processed for the subsequent experiments, including antimicrobial susceptibility testing, PCR, and statistical analysis.

### 4.6. Antimicrobial Susceptibility Testing

All isolates were revived and inoculated onto Müller–Hinton (Oxoid, Basingstoke, Hampshire, UK) plates for antimicrobial susceptibility testing. We determined the resistance of *Salmonella* spp. isolates against a panel of 12 antimicrobials. Antimicrobial susceptibility testing was determined by Kirby–Bauer disk diffusion method according to the Clinical and Laboratory Standards Institute (CLSI). The following antibiotics (Oxoid, Basingstoke, UK; Becton Dickinson, Mississauga, ON, Canada) were used: ampicillin (AMP, 10 µg), amoxicillin-clavulanic acid (AMC, 20/10 µg), chloramphenicol (C 30 µg), gentamicin (CN, 10 µg), tetracycline (TE, 30 µg), trimethoprim-sulfamethoxazole (SXT, 25 µg), erythromycin (E, 15 μg), nalidixic acid (NA, 30 µg), ciprofloxacin (CIP, 5 µg), kanamycin (K, 30 μg), cefoxitin (FOX, 30 µg), and sulphonamides (S, 300 µg). CLSI guidelines were also used to determine breakpoints for classifying isolates as susceptible, intermediate, or resistant to the drug [46]. Multidrug-resistant *Salmonella* spp. was defined as “non-susceptibility to at least one agent in three or more antimicrobial classes” [47]. The multiple antibiotic resistances (MAR) index was determined according to the previously described method [48]. *E. coli* ATCC 25922 was used as the quality control. The breakpoint for resistance or susceptibility interpretation to each antibiotic was in accordance with the CLSI standards. In the evaluation of the results, the strains displaying intermediate resistance were regarded as resistant [49].

### 4.7. DNA Extraction of Salmonella spp. Isolates 

*Salmonella* spp. crude DNA was prepared by using isolated colonies that were sub-cultured overnight in Luria–Bertani broth (Fisher Scientific UK, Loughborough, UK), and genomic DNA was extracted using a Wizard1 Genomic DNA Purification Kit (Promega, Southampton, UK) according to the manufacturer’s instructions. The quality of the extracted DNA was analyzed using spectrophotometer and BE buffer as blank to obtain purified DNA for PCR samples. 

### 4.8. PCR Confirmation of Salmonella spp.

The primers that were used were a genus specific primer for *Salmonella* spp. *invA* gene having the following nucleotide sequence: Forward (5’-3’): GTG AAA TTA TCG CCA CGT TCG GGC AA and Reversed (5’-3’): TCA TCG CAC CGT CAA AGG AAC C. The detailed protocol of the procedure used in this study was performed according to the previously described method [50] and is given in Appendix A.

### 4.9. PCR Assay for Detection of Resistance Genes

The prevalence of genes related to resistance to *bla*_TEM_ for β-Lactams; *tet* (*A*)*,* and *tet* (*B*) for tetracyclines, *cat1, cat2,* and *floR* for chloramphenicol; and *sul1* and *sul2* for sulfonamides was determined by classical PCR. The set of primers used for each gene is shown Appendix A. The primers were designed according to Ye et al. [51]. PCR reactions were performed in a total volume of 25 µL using GoTaq1 Green Master Mix (Promega, Madison, WI, USA), including 12.5 µL of GoTaq1 Green Master Mix, 1 µL of forward primer, 1 µL of reverse primer, 5.5 µL of nuclease-free water, and 5 µL of extracted DNA. Amplification reactions were carried out using a DNA thermocycler (Fisher Scientific UK, Loughborough, UK) according to conditions presented in (Appendix A).

### 4.10. Statistical Analysis

Data were entered into a Microsoft Excel spreadsheet and imported into SPSS version 25 (IBM, Armonk, NY, USA) and the R software (version 3.6.1, https://www.r-project.org/, accessed on 15 January 2021) for statistical analysis. The data were sorted and checked for consistency and duplication. Data visualization was done in ArcGIS v. 10 (esri Inc., Redlands, CA, USA). The data focused on sets of variables that had been previously proposed or identified as risk factors for antimicrobial resistance [52]. Briefly, we classified strains as resistant and not resistant to antimicrobials and then categorized the antimicrobials into their classes then identified which isolates were resistant to one or more specific classes. Classes of antimicrobials included tetracyclines, aminoglycosides, quinolones, sulfonamides, β-Lactams, and chloramphenicol. Prevalence of resistance of *Salmonella* to a panel of 12 antimicrobials was also compared between four different types of samples that included cloacal, faecal, tap water, and sewage samples. Descriptive statistics for frequency of association between AMR and potential risk factors was performed. Selection of variables for inclusion in a logistic regression model was based on prior hypotheses and variables which were suggestive of an important effect from the descriptive analysis.

## Figures and Tables

**Figure 1 pathogens-10-01160-f001:**
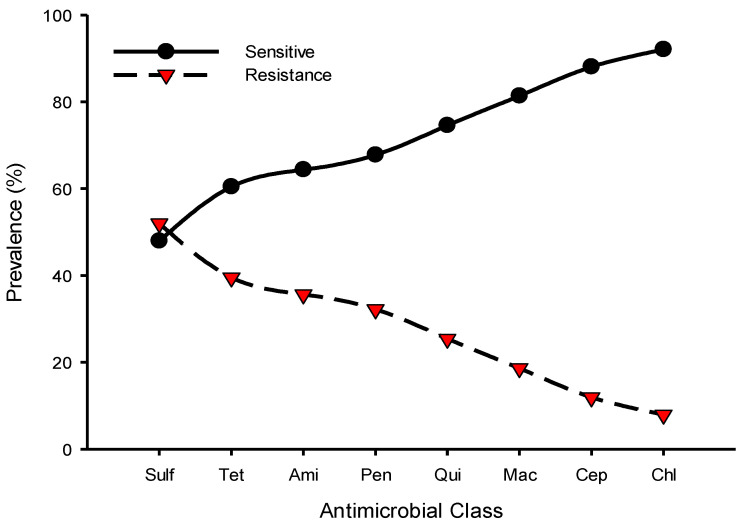
Prevalence of antimicrobial class susceptibility to *Salmonella* spp. isolated from poultry farms collected from Kelantan, Terengganu, and Pahang poultry operations. Data are the number of samples (n = 177). Sulf, sulfonamides; Tet, tetracyclines; Ami, aminoglycosides; Pen, penicillins; Qui, quinolones; Mac, macrolides; Cep, cephalosporins; Chl, chloramphenicol.

**Figure 2 pathogens-10-01160-f002:**
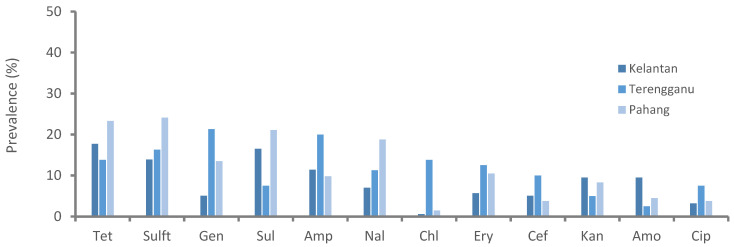
Prevalence of antimicrobial-resistant *Salmonella* spp. isolated from poultry farms collected from Kelantan, Terengganu, and Pahang poultry operations. Data are the number of samples (n = 371). Tet, tetracycline; Sulft, sulfamethoxazole/trimethoprim; Gen, gentamycin; Sul, sulfamethoxazole; Amp, ampicillin; Nal, nalidixic acid; Chl, chloramphenicol; Ery, erythromycin; Cef, cefoxitin; Kan, kanamycin; Amo, amoxicillin; Cip, ciprofloxacin.

**Figure 3 pathogens-10-01160-f003:**
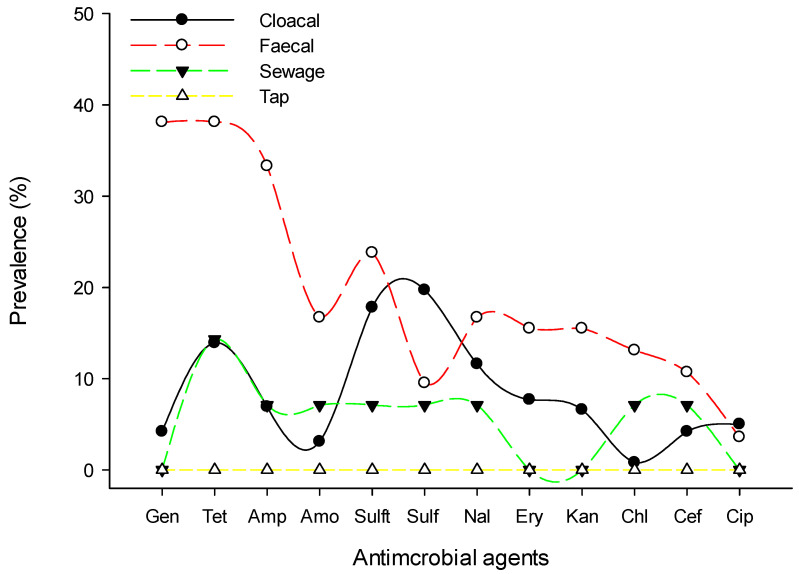
Prevalence of antimicrobial-resistant *Salmonella* spp. by sample collected from poultry farms of East Coast Peninsular Malaysia. Data are the number of poultry strains (n = 371). Gen, gentamycin; Tet, tetracycline; Amp, ampicillin; Amo, amoxicillin; Sulft, trimethoprim-sulphamethoxazole; Sulf, sulfamethoxazole; Nal, nalidixic acid; Ery, erythromycin; Kan, kanamycin; Cef, cefoxitin; Chl, chloramphenicol; Cip, ciprofloxacin.

**Figure 4 pathogens-10-01160-f004:**
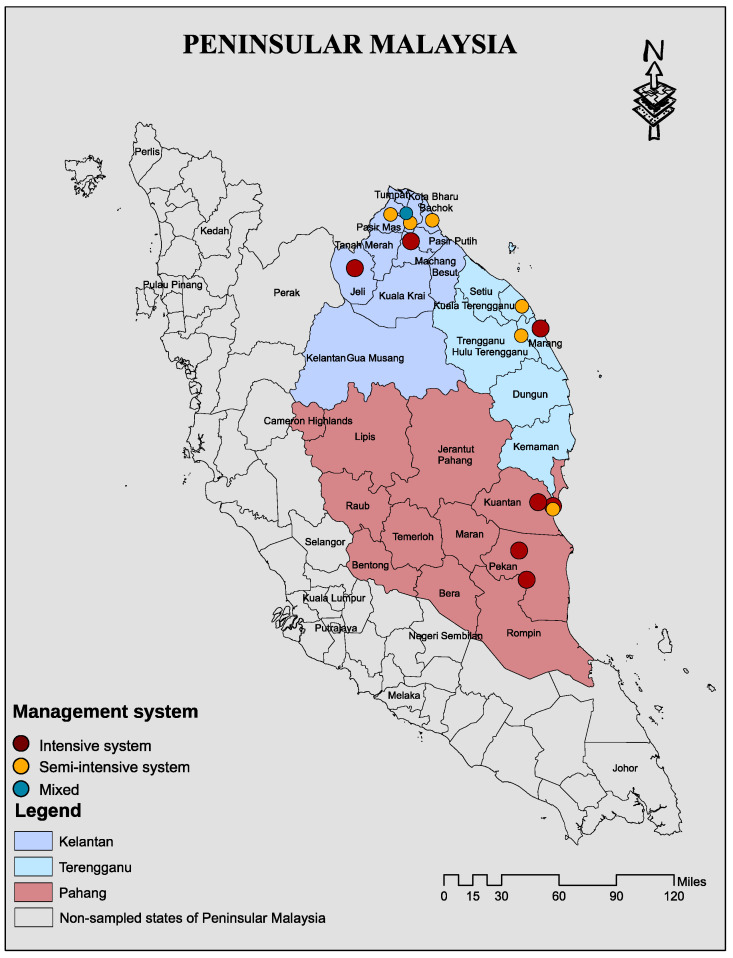
A map showing location of the sampled states and exact location 14 poultry farms sampled and their management systems in Kelantan, Terengganu, and Pahang of East Cost Peninsular Malaysia. The map was created using ArcGIS v. 10 (esri Inc., Redlands, CA, USA).

**Figure 5 pathogens-10-01160-f005:**
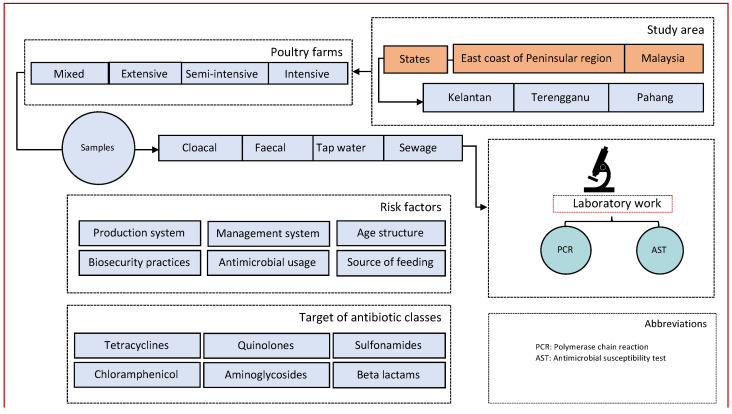
Study organizational flow chart.

**Table 1 pathogens-10-01160-t001:** Summary of risk factors of *Salmonella* spp. among poultry farms in the Kelantan, Terengganu, and Pahang Malaysia (n = 371 samples) by using chi-square analysis.

Risk Factors	Samples Tested	Affected (%)	*p-*Value
Age			0.504
Young	187	86 (46%)	
Adult	184	91 (49.5%)	
Management system			0.478
Intensive	187	95 (50.8%)	
Semi-intensive	158	70 (44.3%)	
Mixed	26	12 (46.2%)	
Production system			0.188
Broiler	212	109 (51.4%)	
Layer	53	25 (47.2%)	
Mixed	106	43 (40.6%)	
State			0.065
Kelantan	158	79 (50%)	
Terengganu	80	29 (36.3%)	
Pahang	133	69 (51.9%)	
Districts			0.010
Kelantan			
Bachok	52	29 (55.8%)	
Kota Bharu	26	12 (46.2%)	
Machang	28	16 (57.1%)	
Pasir Mas	26	13 (50%)	
Jeli	26	9 (34.6%)	
Pahang			
Kuantan	79	50 (63.3%)	
Pekan	54	19 (35.2%)	
Terengganu			
Kuala Terengganu	26	8 (30.8%)	
Marang	54	21 (38.9%)	
Sample source			0.007
Cloaca swab	259	120 (46.3%)	
Fecal Sample	84	50 (59.5%)	
Sewage	14	5 (35.7%)	
Tap Water	14	2 (14.3%)	
Farm size			0.098
Small	104	50 (48.1%)	
Medium	187	97 (51.9%)	
Large	80	30 (37.5%)	
Origin of the poultry			0.113
Local	26	12 (46.2%)	
Imported	133	73 (54.9%)	
Both	212	92 (43.4%)	
Sewage system			0.021
Excellent	109	64 (58.7%)	
Good	210	92 (43.8%)	
Poor	52	21 (40.4%)	
Water Source			0.013
Surface water	106	38 (35.8%)	
Bond water	133	72 (54.1%)	
Pump water	132	67 (50.8%)	

**Table 2 pathogens-10-01160-t002:** Mean of univariate analysis of poultry samples for antimicrobial-resistant *Salmonella* spp. from poultry farms in east coast of Malaysia (n = 177 samples).

Antimicrobial Resistance	Percentage (%)
Resistance	
No resistance	24 (13.6%)
Resistance	153 (86.4%)
Number of classes	
No resistance	24 (13.6%)
Resistant to 1 class	46 (26%)
Resistant to 2 classes	34 (19.2%)
Resistant to 3–4 classes	56 (31.6%)
Resistant to 5 or more classes	17 (9.6%)
Tetracyclines	
Resistant	70 (39.5%)
Penicillins	
Resistant	57 (32.2%)
Aminoglycosides	
Resistant	63 (35.6%)
Sulfonamides	
Resistant	92 (52%)
Cephalosporins	
Resistant	21 (11.9%)
Chloramphenicol	
Resistant	14 (7.9%)
Macrolides	
Resistant	33 (18.6%)
Quinolones	
Resistant	45 (25.4%)

**Table 3 pathogens-10-01160-t003:** Summary of univariate analysis of risk factors for antimicrobial-resistant *Salmonella* spp. from poultry farms in east coast of Malaysia (n = 371 samples).

Risk Factors		Antimicrobials						
		No Identified Resistance		Antimicrobial Class Resistance				
		No Antimicrobial Resistance	Resistance to at Least One Antimicrobial	No Antimicrobial Resistance	Resistant to 1 Class	Resistant to 2 Classes	Resistant to 3–4 Classes	Resistant to 5 or More Classes
Sample type	Cloacal (n = 259)	20 (7.7%)	100 (38.6%)	22 (8.5%)	36 (13.9%)	24 (9.3%)	35 (13.5%)	3 (1.2%)
	Faecal (n = 84)	0	50 (59.5%)	0	7 (8.3%)	9 (10.7%)	21 (34.2%)	13 (15.5%)
	Sewage (n = 14)	2 (14.3%)	3 (21.4%)	2 (14.3%)	2 (14.3%)	0	0	1 (7.1%)
	Tap water (n = 14)	2 (14.3%)	0	2 (14.3%)	0	0	0	0
Age	Young (n = 187)	12 (6.4%)	74 (39.6%)	13 (7%)	20 (10.7%)	13 (7%)	30 (16%)	10 (5.3%)
	Adult (n = 184)	12 (6.5%)	79 (43%)	13 (7.1%)	24 (13%)	21 (11.4%)	26 (14.1%)	7 (3.8%)
Poultry origin	Local (n = 26)	3 (11.5%)	9 (34.6%)	3 (11.5%)	5 (19.2%)	0	4 (15.4%)	0
	Imported (n = 133)	5 (3.8%)	68 (51.1%)	6 (4.5%)	21 (15.8%)	20 (15%)	20 (15%)	6 (4.5%)
	Both (n = 212)	16 (7.5%)	76 (35.8%)	17 (8%)	19 (9%)	13 (6.1%)	32 (15.9%)	11 (5.2%)
Management system	Intensive (n = 187)	7 (3.7%)	88 (47.1%)	9 (4.8%)	21 (11.2%)	18 (9.6%)	34 (18.2%)	13 (7%)
	Semi-intensive (n = 158)	14 (8.9%)	56 (35.4%)	14 (8.9%)	19 (12%)	15 (9.5%)	18 (11.4%)	4 (2.5%)
	Mixed (n = 26)	3 (11.5%)	9 (34.6%)	3 (11.5%)	5 (19.2%)	0	4 (15.4%)	0
Production system	Broiler (n = 212)	13 (6.1%)	96 (45.3%)	15 (7.1%)	26 (12.3%)	20 (9.4%)	36 (17%)	12 (5.7%)
	Layer (n = 53)	1 (1.9%)	24 (45.3%)	1 (1.9%)	9 (17%)	6 (11.3%)	8 (15.1%)	1 (1.9%)
	Mixed (n = 106)	10 (9.4%)	33 (31.1%)	10 (9.4%)	10 (9.4%)	7 (6.6%)	12 (11.3%)	4 (3.8%)
Farm size	Small (n = 104)	9 (8.7%)	41 (39.4%)	9 (8.7%)	18 (17.3%)	8 (7.7%)	13 (12.5%)	2 (1.9%)
	Medium (n = 187)	14 (7.5%)	83 (44.4%)	15 (8%)	21 (11.2%)	19 (10.2%)	33 (17.6%)	9 (4.8%)
	Large (n = 80)	1 (1.3%)	29 (23.8%)	2 (2.5%)	6 (7.5%)	6 (7.5%)	10 (12.5%)	6 (7.5%)
Water source	Surface water (n = 103) Bond water (n = 133)	6 (5.8%) 7 (5.3%)	32 (31.1%) 65 (48.9%)	7 (6.8%) 8 (6%)	3 (2.9%) 20 (15.8%)	5 (4.9%) 16 (12%)	13 (12.6%) 22 (16.5%)	10 (9.7%) 6 (4.5%)
	Pump water (n = 132)	11 (8.3%)	56 (32.2%)	11 (8.3%)	22 (16.7%)	12 (9.1%)	21 (16%)	1 (0.8%)
Sewage system	Excellent (n = 109) Good (n = 210)	4 (3.7%) 16 (7.6%)	60 (55%) 76 (36.2%)	5 (4.6%) 17 (8.1%)	17 (15.6%) 21 (10%)	12 (11%) 19 (9%)	25 (23%) 23 (11%)	5 (4.6%) 12 (5.7%)
	Poor (n = 52)	4 (7.7%)	17 (32.7%)	4 (7.7%)	7 (13.5%)	2 (3.8%)	8 (15.4%)	0
Feed source	Endogenous (n =132)	8 (6%)	53 (40.1%)	8 (6.1%)	20 (15.2%)	14 (10.6%)	16 (12.1%)	3 (2.3%)
	Exogenous (n = 213)	15 (7%)	88 (41.4%)	17 (8%)	19 (9%)	15 (7%)	38 (17.8%)	14 (6.6%)
	Other (n = 26)	1 (3.8%)	12 (46.2%)	1 (3.8%)	6 (23.1%)	4 (15.4%)	2 (7.7)	0

**Table 4 pathogens-10-01160-t004:** Multivariate regression analysis of risk factors for antimicrobial-resistant *Salmonella* spp. from poultry farms in east coast of Malaysia.

	OR	2.5%	97.5%	Pr (>|z|)
Semi-intensive Mixed	Ref	-	-	-
Intensive	1.55	1.01	2.40	0.04
Mixed	0.96	0.39	2.26	0.93

**Table 5 pathogens-10-01160-t005:** Prevalence of *Salmonella* carrying resistance genes.

Antimicrobial Class/Agent	Resistance Gene	% Isolates
Tetracyclines	*tet* (*A*)	7%
Tetracyclines	*tet* (*B*)	14.2%
Chloramphenicol	*cat1*	7%
Chloramphenicol	*cat2*	78%
Chloramphenicol	*floR*	78%
Sulfonamides	*sul1*	85%
Sulfonamides	*sul2*	71%
β-Lactams	*bla* _TEM_	42%

## Data Availability

The data presented in this study are available on request from the corresponding author.

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
