# Peer review of "Antimicrobial Resistance Patterns and Risk Factors Associated with Salmonella spp. Isolates from Poultry Farms in the East Coast of Peninsular Malaysia: A Cross-Sectional Study"

_pathogens, 2021, doi:10.3390/pathogens10091160_

Round 1

Reviewer 1 Report

Paper wrote by Abdinasir Yusuf Osman and co-workes aime to identify the prevalence, resistance patterns, and risk factors associated with
Salmonella isolates from poultry farms. Paper is well wrote, the results are clearly presented and interesting.

Before pubblicantio could be interesting have the syerotype of salmonella isolated to know if the are strains often associated with zoonoses like S. thymurium or infantis.

Moreover recent paper reported the presence of multiresistent strains of salmonella spp in wild boars and others wildlifethis should be added in discussion.

Author Response

RESPONSE TO REVIEWER 1:

We thank the reviewer for reviewing this paper and for the comments and suggestions.  We have tried at our best to revise the manuscript in line with these comments and suggestions.

The specific changes and response to the different points raised include: 

  1. Before publication could be interesting have the serotype of salmonella isolated to know if the are strains often associated with zoonoses like thymurium or infantis. Moreover, recent research reported the presence of multiresistent strains of salmonella spp in wild boars and others wildlife; this should be added in discussion.
  • Response: Thank you for your valuable comments. While serotyping is an interesting and important component of analytical pool of resistance; however, we would like to kindly stress that typing was NOT part of our initial primary research question but we aim to address this in future plans of the forthcoming studies.
  • Regarding the enrichment of the discussion in relation to the presence of multiresistent strains of Salmonella spp. in wild boars and others wildlife; we have added the appended below sentence in the discussion for your refrence. “Generally, poor sewage systems along with presence of manure and rubbish from these operations increase the likelihood of multidrug resistant Salmonella sp. carriage in synanthropic wildlife which in return galvanizes the dissemination of clinically relevant antimicrobial resistance between sympatric wildlife, humans, livestock, and their shared environment. These associations were more pronounced for seed-eating birds and wild boars across different urban ecological systems. This denotes the pressing need to effectively enforce environmental legislation and unregulated antibiotic use in agricultural setting”. (see line 264-270, page 13).

While hoping that these amendments would meet with your favorable consideration, we meanwhile remain completely open to any further suggestions.

Reviewer 2 Report

Generally I highly appreciate the topic of this article as ARB and ARG problematics is of a high importance regarding to the human health. It is an important issue to study the occurrence of different ARGs, specially in food-borne pathogens as Salmonella spp., in the food chain and to detect the critical points for their occurrence and further spreading. I also appreciate the huge effort neccessary for conductig such study.

As the presented data are interesting and the study has a good concept, it requires for their best presentation as possible more detailed explanations of the used methodology which I miss and some improving of the discussion. From this reasons I would like to ask the answers for the following questions to highlight these uncertain parts:

1) Regarding the sampling: At 14 farms 371 samples of 4 different kinds were taken. As the number of sewage samples and tap water (both n = 14) corresponds to the number of farms, it seems that every farm was sampled once for sewage and tap water. For the samples of faeces (n=84) it seems that every farm was sampled by six samples. But for cloacal swabs (n= 259) it is not clear whether the number of samples for one farm was regarding it size or how the number of samples taken from single farms was determined. Please specify this procedure more precisely in 4.4.

2) For 4.5 Microbiological testing it seems you generally followed ISO 6579, but with some limitations (as not using obligatory more selective XLD beside BGA for the isolation from RVS and MKTTn and to use only some of recommended biochemical tests). Regarding this please discuss that using only TSI and lysine iron agar is enough selective to identify only Salmonella spp. Are you sure that no other Enterobacteriaceae species would be able to give the same biochemical profile as Salmonella spp. for these tests ?  Did you do also some other genus confirmation of Salmonella spp. isolates for example by genus specific PCR or immunotesting ? 

The amount of tested samples has to be also specified (it was 10 g or 10 ml for example ?). 

Regarding this Microbiological testing it also seems that finally from every Salmonella spp. positive sample only one isolate was finally chosen for further testing. This is the approach tradionally used in most studies (although it is known that in some sample more strains of Salmonella spp. can be present), but it is neccessary to specify this approach (one positive sample = one isolate in Salmonella spp.) in the methodology.

3) Regarding the PCR detection of ARGs (sul1, cat2, floR, sul2, blaTEM) e.g. on the line 53 it is not clear which isolates were tested for these ARGs - all isolates, only isolates resistant for at least one antibiotic, only isolates resistant to ATB corresponding to these ARGs - e.g. only sulphonamides  resistant antibiotic were tested for the presence of sul1 or sul2 ? Please specify. Was there some strong connection between the phenotypical resistance by disk diffusion method and the presence of ARGs, at least for some group of ATB ?

4) Regarding Table 1 with the conditions for PCR was there some specific reason why for cat1, cat2 and floR the longer time for the initial denaturation (10 minutes instead of 3 minutes for other genes) was used ? All PCR were done by GoTaq1 Green Mastermix, which uses the standard, not hot-start polymerase, requiring the longer time for its activation (although the longer time of the initial denaturaiton should not effect PCR for not hot-start polymerase).

5) Regarding the discussion for the found ATB resistance profile the comparison to other study should be more detailed, not only mentioned as to be the similar (e.g. l. 54-55). It could be discussed although in the connection to the most important trends in other countries as e.g. for the resistance to ciprofloxacine which is in many countries more higher than in this study. As this could be connected to the spectrum and practice of ATB usage, it could be also reflected to some level in the discussion.

6) In 4.3 it is specified that the samples collecting was performed within one year from Feb 2019 and Feb 2020. The surviving and multiplying of Salmonella spp. in chicken farms could be effected also by the environmental conditions as the temperature and humidity. Did not you observe that the climatic factor should be taken as another risk factor or is it its impact negligable ? 

7) Regarding the formal issues:

a) in many places the word Salmonella spp. is incorrectly written (without italics in l. 101-106, Salmonella Spp., in l. 26 etc.),

b) in Table 1 the additional empty line should be added before the general name as Production system, State etc.,

c) in Table 2 - is it really neccessary to duplicate data as expressing them both Sensitive/Resistant which is always summed as 100 % ? was not expressing only Resistant enough ?

d) I recommend to post Table 3 in Supplementary materials as it only supports the analysis in Table 4

e) For Figure 4 I recommend to improve using the colours (e.g. to use white colour for the map outside the tested district) - in the actual colour arragmnet it doesnot provide an easy survey.

f) In Figure 2 the description of the second column is incorrect.

Author Response

RESPONSE TO REVIEWER 2:

We thank the reviewer for reviewing this paper and for the comments and suggestions.  We have tried at our best to revise the manuscript in line with these comments and suggestions.

The specific changes and response to the different points raised include: 

  1. Regarding the sampling: At 14 farms 371 samples of 4 different types of samples were taken. As the number of sewage samples and tap water (both n = 14) corresponds to the number of farms, it seems that every farm was sampled once for sewage and tap water. For the samples of faeces (n=84) it seems that every farm was sampled by six samples. But for cloacal swabs (n= 259) it is not clear whether the number of samples for one farm was regarding it size or how the number of samples taken from single farm was determined. Please specify this procedure more precisely in 4.4.

  • Response: Thank you for this comment. In response, we have now given additional table as supplementary material indicating the share of samples per farm for further clarification (Table S2).

  • Table S2: Share of samples per farm in east coast of peninsular Malaysia

Farms

Cloacal swabs

Faecal

Sewage

Tap water

Farm 1

20

6

1

1

Farm 2

18

6

1

1

Farm 3

18

6

1

1

Farm 4

18

6

1

1

Farm 5

18

6

1

1

Farm 6

18

6

1

1

Farm 7

20

5

1

1

Farm 8

18

7

1

1

Farm 9

18

6

1

1

Farm 10

18

6

1

1

Farm 11

19

6

1

1

Farm 12

18

6

1

1

Farm 13

18

6

1

1

Farm 14

20

6

1

1

Total

259

84

14

14

  1. For 4.5 Microbiological testing it seems you generally followed ISO 6579, but with some limitations (as not using obligatory more selective XLD beside BGA for the isolation from RVS and MKTTn and to use only some of recommended biochemical tests). Regarding this please discuss that using only TSI and lysine iron agar is enough selective to identify only Salmonella Are you sure that no other Enterobacteriaceae species would be able to give the same biochemical profile as Salmonellaspp. for these tests?  Did you do also some other genus confirmation of Salmonella spp. isolates for example by genus specific PCR or immunotesting? The amount of tested samples has to be also specified (it was 10 g or 10 ml for example?). Regarding this Microbiological testing it also seems that finally from every Salmonella spp. positive sample only one isolate was finally chosen for further testing. This is the approach tradionally used in most studies (although it is known that in some sample more strains of Salmonella spp. can be present), but it is neccessary to specify this approach (one positive sample = one isolate in Salmonella spp.) in the methodology.
  • Response: Thank you for this comment. Using TSI and Lysine iron is not selective only for Salmonella spp. However, characteristics round transparent colonies with black centers on XLD and pink to red colonies with reddening of BGA is characteristics of Salmonella. Additionally, we were careful not to include the serotypes name because we did not serotype the isolates as this was outside the objectives of the research work.

  1. Regarding the PCR detection of ARGs (sul1cat2floRsul2blaTEM) e.g. on the line 53 it is not clear which isolates were tested for these ARGs - all isolates, only isolates resistant for at least one antibiotic, only isolates resistant to ATB corresponding to these ARGs - e.g. only sulphonamides  resistant antibiotic were tested for the presence of sul1or sul2 ? Please specify. Was there some strong connection between the phenotypical resistance by disk diffusion method and the presence of ARGs, at least for some group of ATB?
  • Response: Only resistant isolates that corresponds to the antibiotics were assayed for the presence of antimicrobial resistance genes. Yes, there was a correlation between carriage of ARGs and phenotypic resistance using disk diffusion.

  1. Regarding Table 1 with the conditions for PCR was there some specific reason why for cat1cat2and floR the longer time for the initial denaturation (10 minutes instead of 3 minutes for other genes) was used? All PCR were done by GoTaq1 Green Mastermix, which uses the standard, not hot-start polymerase, requiring the longer time for its activation (although the longer time of the initial denaturaiton should not effect PCR for not hot-start polymerase).

  • Response: There is no specific reason from our end, the primers and PCR cycling conditions were as previously in [46]. We used the same primers and after optimizing the protocol, it worked well for us.

  1. Regarding the discussion for the found ATB resistance profile the comparison to other study should be more detailed, not only mentioned as to be the similar (e.g. l. 54-55). It could be discussed although in the connection to the most important trends in other countries as e.g. for the resistance to ciprofloxacine which is in many countries more higher than in this study. As this could be connected to the spectrum and practice of ATB usage, it could be also reflected to some level in the discussion.

  • Response: We have now discussed in detail and made comparison in connection to the most important and relevant trends. (see line 210-215, page 12; see line 219-22, page 12 ). In more detail, we have added  the following sentence:

  • ‘For example, chicken flock sampling in South-central Peninsular  foundMalaysia found that Salmonella spp. resistant to ampicillin (17.6%), tetracycline and streptomycin (35.3%), Sulfonamides (29.4%), trimethoprim (20.6%) nalidixic acid and Colistin (14.7%), chlo-ramphenicol and Nitrofurantoin (11.7%), Amoxicillin-Clavulanate(5.9%), Kanamycin and Cefotaxime (2.9%), gentamicin, Ciprofloxacin, Norfloxacin and Ceftiofur (0%) (see line 210-215, page 12).
  • Similar discussion were also noted in (see line 219-22, page 12)

  • ‘For example, Salmonella spp. isolated from poultry and environmental samples collected from wet markets in the northern region of peninsular Malaysia were highly resistant to sulphonamide (96.5%), ampicillin (89.5%), tetracycline (85.1%), chloramphenicol (75.4%), trimethoprim (68.4%), trimethoprim-sulfamethoxazole (67.5%), streptomycin (58.8%) and nalidixic acid (44.4%)’ (see line 219-22, page 12).

  1. In 4.3 it is specified that the samples collecting was performed within one year from Feb 2019 and Feb 2020. The surviving and multiplying of Salmonella  in chicken farms could be effected also by the environmental conditions as the temperature and humidity. Did not you observe that the climatic factor should be taken as another risk factor or is it its impact negligible? 

  • Response: Thank you for your valuable comments. Indeed the effect of the climatic factor on the spread of antimicrobial resistance is NOT negligible and little is known about the effect of climate on the prevalence of antimicrobial resistance in Malaysia. However, our study was just snapshot and we have NOT investigated climatic variation which indeed require subnational data and we aim this to capture in the future plans of subsequent analysis.

  1. Regarding the formal issues: (a) In many places the word Salmonella  is incorrectly written (without italics in l. 101-106, SalmonellaSpp., in l. 26 etc.)
  • Response: Thank you for bringing this to our attention. We have now italicized the ‘Salmonella spp.’ In the entire manuscript (see the track version [Pathogens-1328074_R_Not clean]).

(b)in Table 1 the additional empty line should be added before the general name as Production system, State etc.,

  • Response: We have now deleted the additional line in Table 1. (see line 160-161, page 6).

(c) In Table 2 - is it really necessary to duplicate data as expressing them both Sensitive/Resistant which is always summed as 100 %? Was not expressing only Resistant enough?

  • Response: Thank you for your helpful suggestion. We have now deleted the rows corresponding to sensitive and retained those with resistance to avoid duplication. (see line 163-164, page 7-8).

(d) I recommend posting Table 3 in Supplementary materials as it only supports the analysis in Table 4.

  • Response: Thank you for your recommendation. We have now removed Table 3 from the manuscript and posted in supplementary materials accordingly (see line 164-165, page 8).

(e) For Figure 4, I recommend to improve using the colours (e.g. to use white colour for the map outside the tested district) - in the actual colour arrangement it does not provide an easy survey.

  • Response:  Thank you for this comment. The colours of the non-sampled areas were already in white, but the background is NOT. Now, we have added additional layer in the legend indicating the non-sampled area for further clarification to the reader (see line 294-295, page 15).

(f) In Figure 2, the description of the second column is incorrect.

  • Response: Thank you for your bringing this to our attention. We have now re-plotted the figure and rectified the description of the 2nd column (see line 148, page 5)

OTHER:

Abstract:

The burden of antimicrobial use in agricultural settings is one of the greatest challenges facing global health and food security in the modern era. Malaysian poultry operations are relevant but understudied component of epidemiology of antimicrobial resistance. (see line 21-23, page 1).

While hoping that these amendments would meet with your favorable consideration, we meanwhile remain completely open to any further suggestions.

***